# Scaling MLPs: A Tale of Inductive Bias

**Gregor Bachmann**[*], **Sotiris Anagnostidis**[*], **Thomas Hofmann**
ETH Zürich, Switzerland

## Abstract

In this work we revisit the most fundamental building block in deep learning, the multi-layer perceptron (MLP), and study the limits of its performance on vision tasks. Empirical insights into MLPs are important for multiple reasons. (1) Given the recent narrative *"less inductive bias is better"*, popularized due to transformers eclipsing convolutional models, it is natural to explore the limits of this hypothesis. To that end, MLPs offer an ideal test bed, as they lack any vision-specific inductive bias. (2) MLPs have almost exclusively been the main protagonist in the deep learning theory literature due to their mathematical simplicity, serving as a proxy to explain empirical phenomena observed for more complex architectures. Surprisingly, experimental datapoints for MLPs are very difficult to find in the literature, especially when coupled with large pre-training protocols. This discrepancy between practice and theory is worrying: *Do MLPs reflect the empirical advances exhibited by practical models?* Or do theorists need to rethink the role of MLPs as a proxy? We provide insights into both these aspects. We show that the performance of MLPs drastically improves with scale (95% on CIFAR10, 82% on CIFAR100, 58% on ImageNet ReaL), highlighting that lack of inductive bias can indeed be compensated. We observe that MLPs mimic the behaviour of their modern counterparts faithfully, with some components in the learning setting however exhibiting stronger or unexpected behaviours. Due to their inherent computational efficiency, large pre-training experiments become more accessible for academic researchers. All of our experiments were run on a single GPU.

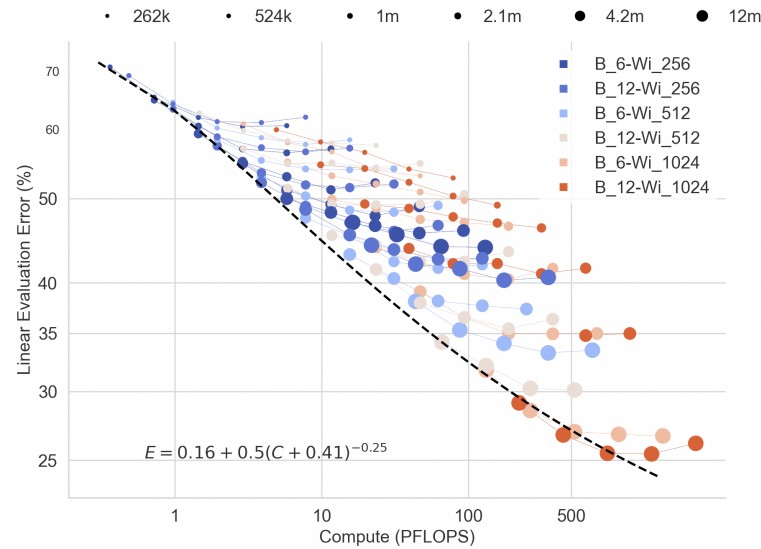

Figure 1: Test error on CIFAR100 as a function of PFLOPS.

---

[*]Equal contribution. Correspondence to {`gregorb, sanagnos`}@ethz.ch. Code and checkpoints available at https://github.com/gregorbachmann/scaling_mlps

37th Conference on Neural Information Processing Systems (NeurIPS 2023).

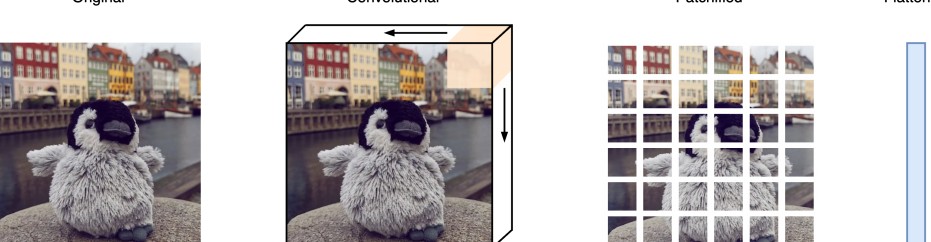

Figure 2: Different architectures process images differently. Convolutions directly operate on the image, ViTs and MLP-Mixers work with patches while the MLP takes the flattened image as input.

# 1  Introduction

Deep learning has undergone tremendous empirical progress in the last decades. The dominant approaches in practice these days rely on very large, pre-trained models which are then fine-tuned to the specific task at hand. For natural language processing, these models usually are some variant of the Transformer architecture (Vaswani et al., 2017), while in computer vision, both convolutional and transformer-based models are very popular (He et al., 2015; Tan and Le, 2020; Dosovitskiy et al., 2021). The theoretical understanding of these advances on the other hand remains very poor and the gap between the world of theory and practice is growing at an alarming rate. One aspect of this gap is the family of models investigated; due to their mathematical simplicity, theoretical works largely focus on simple multi-layer perceptrons (MLPs). Consisting of a series of unstructured matrix multiplications, interleaved with element-wise non-linearities, the MLP serves as an ideal test bed to analyze empirical phenomena exhibited by more complicated models employed in practice. Due to their inferior performance, MLPs are rarely used and very little is known regarding their behaviour in more modern settings. For instance, to the best of our knowledge, there is not a single published result showcasing an MLP trained on ImageNet1k, the de-facto standard benchmark in vision, let alone any pre-training/transfer learning studies. This lack of empirical data is concerning as theory aims to understand the characteristics of modern architectures through the lens of MLPs, yet only little assessments are made regarding how well such a proxy works. This raises the question,

$$\textit{Do MLPs reflect the empirical advances exhibited by practical models?} \qquad (1)$$

Investigating MLPs is not only interesting for theory but also for practice. With the Vision Transformer (ViT) outperforming its convolutional competitors in very large-scale settings, the role of inductive bias has recently been brought into question. Since a ViT is equipped with significantly less inductive bias for vision compared to convolutional models (e.g. it lacks translation-equivariance) a novel narrative has recently emerged:

$$\textit{At large scales of compute, having less inductive bias is beneficial for performance.} \qquad (2)$$

More evidence for this hypothesis has been collected in the form of the MLP-Mixer (Tolstikhin et al., 2021), an architecture with arguably even less inductive bias, solely relying on multi-layer perceptrons as patch processors and mixers. The MLP architecture is the ideal candidate to test the limits of such a hypothesis, as it exhibits the least inductive bias for vision due to its invariance to permutations of pixels. Unfortunately, the scale where Transformers and MLP-Mixers start to outperform convolutional models is out of reach for most researchers, requiring billions of annotated images and thousands of TPUs. We thus expect similar required scales for MLPs and hence instead investigate the following, weaker hypothesis:

$$\textit{Lack of inductive bias can be compensated by scaling compute.} \qquad (3)$$

i.e. we aim to measure to what degree a lack of inductive bias hinders performance even if a model is subjected to a large parameter count and trained on datasets with many examples (albeit smaller than what is employed in Dosovitskiy et al. (2021)).

In this work, we provide answers to question 1 and provide further evidence for hypothesis 2 and 3 by investigating how far we can push the empirical performance of models solely built from composing several MLP blocks. We give largely positive answers to question 1, observing that MLPs behave very similarly to their modern counterparts when subjected to scale, i.e. their performance increases

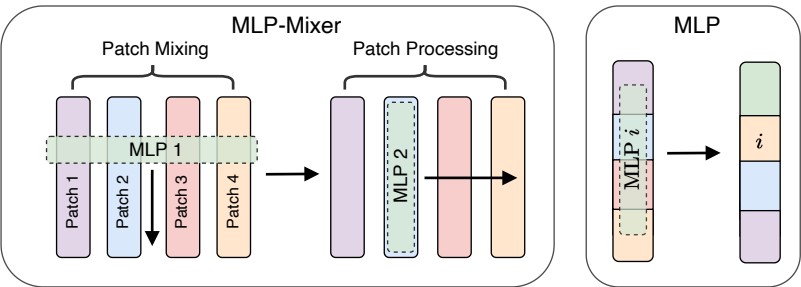

Figure 3: A simplified depiction of the differences between an MLP-Mixer and an MLP.

predictably as a power law in parameter count and sample size, akin to Hestness et al. (2017, 2019); Kaplan et al. (2020); Zhai et al. (2022) (see e.g. Fig. 1). In contrast to previous work, however, we find that compute-optimal MLPs allocate their budget more strongly into sample size, highlighting again their small inductive bias. While regularization in the form of data augmentation is also helpful for CNNs, its role is significantly amplified for MLPs even at large sample sizes, leading to fatal degradation if turned off. We further investigate how the implicit bias of SGD affects performance, and we make a very counter-intuitive discovery: contrary to CNNs, larger batch sizes generalize significantly better for MLPs. This result questions the validity of the proxy role that the MLP plays in theoretical works investigating the implicit bias of SGD. While, as expected, the scale employed in this work does not suffice for hypothesis 2, we provide strong evidence for 3, which we view as an important first step. We observe that scale indeed suffices to overcome the bad inductive bias present in MLPs, leading to surprisingly strong downstream performance, e.g. $\approx 95\%$ on CIFAR10, $\approx 82\%$ on CIFAR100 and $\approx 58\%$ on ImageNet ReaL. In summary, we make the following contributions:

- We fill the gap between theory and practice, providing the first results for MLPs trained in modern settings.

- We show that MLPs mostly behave comparably to their modern counterparts, making them a good proxy for theory. We observe however that the roles of regularization and implicit bias of SGD significantly differ and theory hence needs to adapt.

- We provide further evidence that inductive bias is not crucial at large scales, showing that even "bad" architectures like MLPs can achieve strong downstream performance. We however identify a shift in compute-optimality, showing that optimal MLPs invest their compute significantly more into dataset size compared to model size.

## 2 Background

**Theoretical Works.** The MLP has served as the main object of study for theoretical works in deep learning across different domains. The cornerstone results for areas such as convergence of SGD-trained neural networks (Mei et al., 2018; Du et al., 2019; Zou et al., 2020; Li and Yuan, 2017; Saxe et al., 2014), most generalization bounds (Arora et al., 2019b; Mei and Montanari, 2021; Jacot et al., 2018; Allen-Zhu et al., 2019a), the benefits of overparametrization (Neyshabur et al., 2019; Allen-Zhu et al., 2019b; Arora et al., 2018), the implicit bias of SGD towards favourable solutions (Soudry et al., 2018; Neyshabur et al., 2014; Chizat and Bach, 2020), signal propagation properties (Poole et al., 2016; Schoenholz et al., 2017) and scaling laws (Bahri et al., 2021; Maloney et al., 2022) are all largely obtained for MLPs. To quote the very influential *Principles of Deep Learning Theory* book (Roberts et al., 2022):

*"MLPs are the simplest of these neural network architectures that hinge on this stacking idea, and thus provide a minimal model for an effective theory of deep learning."*

There are also several theoretical works studying more modern setups such as convolutional or transformer-based networks including Arora et al. (2019a); Gunasekar et al. (2018); Brutzkus and Globerson (2017); Hron et al. (2020) to name but a few, but the main theoretical focus to the best of our knowledge still remains on the MLP architecture. We thus believe it is important to explore the limits of such a theoretical proxy in realistic settings.

**MLPs.** The multi-layer perceptron has its origins in Rosenblatt (1958), serving as an extension to the classic Perceptron with its hidden layers however fixed to random initialization. Ivakhnenko et al. (1965) devised the first method to update the hidden layers through *self-organization*. Amari (1967) then introduced the idea to train the parameters with stochastic gradient descent. Mathematically, an MLP of depth $L \in \mathbb{N}$ can be described very efficiently; given an input $\boldsymbol{x} \in \mathbb{R}^d$, it applies a series of linear transformations, interleaved with an element-wise non-linearity $\sigma : \mathbb{R} \to \mathbb{R}$:

$$\boldsymbol{z}^{(l)} = \boldsymbol{W}^{(l)} \boldsymbol{x}^{(l-1)} \quad \longrightarrow \quad \boldsymbol{x}^{(l)} = \sigma\left(\boldsymbol{z}^{(l)}\right)$$

where we define $\boldsymbol{x}^{(0)} := \boldsymbol{x}$ and $\boldsymbol{W}^{(l)} \in \mathbb{R}^{d_l \times d_{l-1}}$ for $l = 1, \dots, L$ are the learnable weight matrices. For the sake of readability, we omit the biases. This mathematical simplicity makes the MLP a very attractive model to study theoretically (albeit still very far from trivial) and indeed many works frame their results around this general model class. When used for vision, the input tensor $\boldsymbol{x} \in \mathbb{R}^{h \times w \times 3}$ is flattened into a vector $\mathrm{vec}(\boldsymbol{x}) \in \mathbb{R}^{3hw}$. Notice how such an architecture completely lacks locality and weight sharing, every unit simply processes the entire image at once. More worryingly, the vectorization $\mathrm{vec}$ could be applied in any way, i.e. any permutation of $\boldsymbol{x}$ looks identical to an MLP.

We want to highlight that MLPs of course are not completely free of inductive bias, in the sense that they encourage learning a hierarchical feature structure. On the other hand, there is no vision-specific inductive bias present in MLPs, which is the main setting we investigate here. We refer to Battaglia et al. (2018) for a more in-depth treatment of inductive bias.

**Convolutions.** The MLP is a very general model and has no structure built into it to make it more suitable for vision tasks. A convolution on the other hand was designed specifically for vision with desirable characteristics incorporated into the model. A convolution can be viewed as a special case of an MLP, where the weight matrix $\boldsymbol{W}$ is very structured by being sparse and having shared entries, leading to spatially localized learning. This can be most easily illustrated in the case of convolving a $2 \times 3 \times 1$ image $\boldsymbol{x}$ with a $2 \times 2$ filter $\boldsymbol{f}$ as the following matrix multiplication:

$$\boldsymbol{f} * \boldsymbol{x} = \boldsymbol{W_f} \, \mathrm{vec}(\boldsymbol{x}) = \begin{pmatrix} f_1 & f_2 & 0 & f_3 & f_4 & 0 \\ 0 & f_1 & f_2 & 0 & f_3 & f_4 \end{pmatrix} \mathrm{vec}(\boldsymbol{x})$$

Here $\mathrm{vec}$ denotes the standard, row-wise vectorization-scheme to flatten the image. Instead of operating with a dense matrix as the MLP, the convolution uses a structured matrix $\boldsymbol{W_f}$ tailored to the task of vision, leading to a better inductive bias. Moreover, a convolution exhibits translation-equivariance, i.e. shifts of images are processed equivalently to the original. Crucially, in contrast to the MLP, a convolution severely suffers if a permutation is applied to the image.

**Vision Transformer.** Inspired by successes in NLP, the Transformer architecture has been adapted to vision (Dosovitskiy et al., 2021). An image $\boldsymbol{x} \in \mathbb{R}^{h \times w \times 3}$ is broken up into patches (also called tokens) and linearly embedded (see Fig. 2), augmented with a positional embedding, marking its spatial location in the image. The obtained embeddings are processed by self-attention layers where patches interact, and MLP layers, which are shared among patches and transform them individually. While the inductive bias of a ViT is certainly weaker compared to a CNN (it lacks translation-equivariance), the patching and parameter sharing still make the architecture suitable for vision.

**MLP-Mixer.** Similar to the ViT, the MLP-Mixer also works with a patchified image (Tolstikhin et al., 2021). Unlike the ViT, token-mixing is not implemented using self-attention but rather another MLP block is used to exchange information between patches. We want to clearly highlight the difference between an MLP-Mixer and an MLP: An MLP-Mixer operates on patches, where in each block it applies a shared MLP to each patch for processing, and another MLP for mixing the patches along the channels. We visualize the differences in Fig. 3 for clarity. We again want to stress that breaking the image into patches and sharing parameters among them significantly enhances the amount of inductive bias, compared to a standard MLP.

**Patchifiying.** As highlighted above, ViTs and Mixers largely obtain their inductive biases through breaking the images into patches. This choice seems to be beneficial even for architectures that already possess a strong inductive bias, such as the ConvMixer (Trockman and Kolter, 2022), where convolutions are performed on individual patches. The very recent Metaformer (Yu et al., 2022) further shows that even a simple spatial pooling instead of attention can lead to strong performance if the image is patchified. While the success of this mechanism certainly warrants further investigation, in this work we decided to deliberately focus on MLPs as they specifically lack this type of bias.

|  | CIFAR10 | CIFAR100 | TINYIMAGENET | IMAGENET |
|---|---|---|---|---|
| S-MLP (@100 E) | 54.2 | 28.8 | 8.5 | 9.2 |
| S-MLP + DA (@ 1000 E) | 68.9 | 43.3 | 25.2 | 24.3 |
| S-MLP + DA (@ 5000 E) | 72.3 | 44.5 | 27.3 | 26.8 |
| B-MLP (@ 100 E) | 58.1 | 30.5 | 8.9 | 8.7 |
| B-MLP + DA (@1000 E) | 70.1 | 48.3 | 27.2 | 28.7 |
| B-MLP + DA (@5000 E) | 75.4 | 50.4 | 31.2 | 31.7 |
| RESNET18[2] + DA | 93.2 | 75.6 | 68.9 | 69.7 |

Table 1: Test accuracies (in %) without any pre-training. The S-MLP has depth 6 and width 1024 while the B-MLP has depth 6, width 1024 and an expansion factor of 4.

## 3 Architecture

We study different variants of the MLP architecture, starting from the standard vanilla setup and then adding more components such as residual connections and bottleneck layers.

**Standard MLP.** As a first starting point, we investigate simple MLPs with ReLU activations and isotropic design, i.e. except for the first, every layer has the same width $m \in \mathbb{N}$. In order to avoid training instabilities we further enhance the standard MLP with layer normalizations (Ba et al., 2016) placed after the activations. We thus compose several blocks of the form

$$\text{Block}(\boldsymbol{z}) = \sigma\left(\boldsymbol{W}\,\text{LN}(\boldsymbol{z})\right)$$

with $\boldsymbol{W} \in \mathbb{R}^{m \times m}$. To embed the image $\boldsymbol{x} \in \mathbb{R}^{d \times d \times 3}$ we use a linear layer $\text{emb}(\boldsymbol{x}) = \boldsymbol{W}^{emb}\,\text{vec}(\boldsymbol{x})$ with $\boldsymbol{W}^{emb} \in \mathbb{R}^{m \times 3d^2}$. Such an embedding layer is crucial since for high resolution images, $3d^2$ can be quite large and thus $m$ needs to be chosen smaller. We empirically find that such a network design is the minimal choice in order to guarantee successful training across all scales of parameter count and sample size. We will use the short cut *S-MLP* to denote such an architecture.

**Inverted Bottleneck MLP.** Inspired by Lin et al. (2015); Tolstikhin et al. (2021) we add a bottleneck structure to an MLP block as well as skip connections as follows:

$$\text{Block}(\boldsymbol{z}) = \boldsymbol{z} + \boldsymbol{W}^c \sigma\left(\boldsymbol{W}^e\,\text{LN}\left(\boldsymbol{z}\right)\right)$$

where $\boldsymbol{W}^e \in \mathbb{R}^{km \times m}$ expands the dimension to $km$ for $k \in \mathbb{N}$ and $\boldsymbol{W}^{(c)} \in \mathbb{R}^{m \times km}$ collapses it back to width $m$. For most experiments we set $k = 4$. While the additions of skip connections and bottleneck layers to the architecture arguably add some amount of inductive bias, we believe that in comparison to modern architectures such enhancements remain negligible. We will denote this variant by *B-MLP*.

## 4 Experiments

**Setup** In this work, we solely focus on vision tasks as inductive bias is more readily understood in this setting. Moreover, most theoretical works focus on image classification tasks, making it thus a natural test bed to assess the performance of MLPs. We study the popular tasks CIFAR10, CIFAR100 (Krizhevsky, 2009), STL10 (Coates et al., 2011), TinyImageNet (Le and Yang, 2015), ImageNet1k for evaluation, as well as ImageNet21k (Deng et al., 2009) for pre-training. To limit the size of the embedding layer and the computational needs, we downscale images to resolution $64 \times 64 \times 3$ (if needed) as done in Chrabaszcz et al. (2017). We center and normalize all the images and use random flips and crops as well as MixUp (Zhang et al., 2018) as data augmentations.

---

[2]In contrast to the MLPs, the ResNet18 was trained at the original image resolutions.

|  | CIFAR10 | CIFAR100 | STL10 | TINY-IN | IN | REAL |
|---|---|---|---|---|---|---|
| *B-6/Wi-1024* | $69.9_{\pm0.1}$ | $43.0_{\pm0.4}$ | $51.5_{\pm0.1}$ | $47.1_{\pm0.1}$ | $15.2_{\pm0.2}$ | $20.3_{\pm0.2}$ |
| *B-6/Wi-1024* + DA | $91.5_{\pm0.02}$ | $76.4_{\pm0.2}$ | $85.0_{\pm0.2}$ | $62.7_{\pm0.1}$ | $38.7_{\pm0.1}$ | $47.0_{\pm0.15}$ |
| *B-12/Wi-1024* + DA | $94.2_{\pm0.05}$ | $80.0_{\pm0.05}$ | $89.9_{\pm0.1}$ | $69.9_{\pm0.4}$ | $43.3_{\pm0.06}$ | $48.6_{\pm0.2}$ |
| *B-12/Wi-1024* + DA + TTA | $95.5_{\pm0.05}$ | $82.6_{\pm0.2}$ | $92.2_{\pm0.05}$ | $73.1_{\pm0.5}$ | $51.5_{\pm0.1}$ | $57.9_{\pm0.1}$ |

Table 2: Fine-tuning Top-1 accuracies (in %) when pretrained on ImageNet21k. Accuracies are averaged over 3 runs. For readability, we abbreviate ImageNet as IN.

## 4.1 Training from Scratch

We start the empirical exploration of MLPs by training them from scratch (i.e. without any extra data) on popular vision benchmarks. All models were trained with the LION optimizer (Chen et al., 2023) with a learning rate $\eta = 5e\text{-}5$. In order to combat overfitting we use strong label smoothing $\alpha = 0.3$. We display the resulting test accuracies in Table 1. We observe that both the standard architecture and the bottleneck without any data augmentation suffer from overfitting, leading to suboptimal performance. When turning it on, data augmentation as a regularizer however really unfolds its full power, significantly pushing the performance by roughly 20% across all tasks. As observed in Lin et al. (2015), the inverted bottleneck architecture leads to an improvement in performance across all datasets. Learning on the other hand significantly slows down with strong augmentations such as MixUp, enabling training for up to 5000 epochs without suffering from overfitting. However, compared to simple modern baselines such as a ResNet18 (He et al., 2015), a large discrepancy in performance remains, highlighting the importance of inductive bias in the small sample regime. We remark that ViTs and MLP-Mixers as well exhibit more learning difficulties if the dataset size is small (Dosovitskiy et al., 2021; Tolstikhin et al., 2021). We provide more ablation studies in Appendix A.2.

## 4.2 Transfer Learning

In this section, we aim to analyze how transferable features learnt by MLPs are across different vision tasks. Transferability is one of the hallmark characteristics of modern deep learning, enabling practitioners to fine-tune large models on their specific dataset, leading to superior performance. We are, to the best of our knowledge, the first to measure transferability of MLPs, which is crucial to assess in order to build a theoretical understanding of the process. In this section, we focus on the inverted bottleneck MLP as it generalizes better and is easier to optimize. We provide the dual results for the standard MLP in Appendix B.1. We restrict to $k = 4$ for the expansion factor and denote by *B-L/Wi-m* a network with $L$ blocks and width $m$. For pre-training we use ImageNet21k, the largest publicly available image dataset with annotated classes. After preprocessing the dataset following Ridnik et al. (2021), it consists of roughly 12 million images and 11 thousand classes. We then pre-train the MLP with the cross-entropy loss for 800 epochs, employing label smoothing and the LION optimizer. To guarantee fast data loading we rely on the FFCV framework (Leclerc et al., 2023) for all experiments.

In order to measure transferability of the learnt features we fine-tune the network on the new task. We also study training a linear layer on top of the embeddings but defer those results to Appendix A.3. We again explore the effects of data augmentation during the pre-training stage. For fine-tuning we use SGD with momentum with a learning rate of $\eta_{\text{head}} = 0.01$ for the head and $\eta_{\text{body}} = 0.001$ for the encoder for 50 epochs. We upscale CIFAR images to resolution $64 \times 64 \times 3$ at fine-tuning time to guarantee compatibility. We display the fine-tuning results in Table 2. For visualizations of the learnt features, we refer the interested reader to Appendix C. We again observe that using data augmentation during the pre-training phase is essential to successful training, boosting performance up to 30% in case of CIFAR100. Surprisingly, the learnt features are highly transferable, improving the performances reported previously in Table 1 dramatically. While of course pre-trained on a large quantity of data, we nevertheless want to highlight that such an MLP becomes competitive with a ResNet18 trained from scratch for all the datasets, except for ImageNet1k where performance falls surprisingly short. We hypothesize that MLPs struggle with the more fine-grained distinctions between classes, in combination with the reduced resolution of the images.

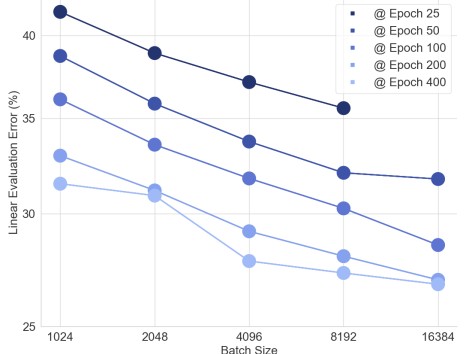

| Model | #parameters |
|-------|-------------|
| *B-6/Wi-256* | 9M |
| *B-12/Wi-256* | 12M |
| *B-6/Wi-512* | 24M |
| *B-12/Wi-512* | 37M |
| *B-6/Wi-1024* | 74M |
| *B-12/Wi-1024* | 124M |

Figure 4: Linear downstream error on CIFAR100 (in %) when pretrained for varying batch-sizes on ImageNet21k, on a log-log scale.

Table 3: The different models and the respective parameter counts in millions.

**Test-Time Augmentations.** For ImageNet1k we further notice that objects tend to not be centered, in contrast to datasets like CIFAR10. We suspect that this might lead to the comparatively weaker performance. To test this, we leverage test-time augmentations (TTA). As introduced by Krizhevsky et al. (2012), for each test image, we produce a fixed number of 100 random crops and use the averaged logits for prediction. We observe significant improvements across all datasets, especially for ImageNet we obtain an increase of roughly 8%. This indeed indicates that MLPs struggle to localize the object of interest, especially for the more complicated ImageNet1k task. Using a large number of crops alleviates this problem to some degree. This also explains why the gains on tasks like CIFAR10 are smaller as the objects there usually are perfectly centered.

**ReaL accuary.** As observed in (Beyer et al., 2020), the ImageNet labels do not capture that a single image might contain multiple objects of distinct classes. ImageNet accuracy can thus be misleading in the sense that model classes such as convolutional networks might have implicitly adapted to the particular labeling strategy due to the repeated benchmarking on the same validation set. MLPs most likely lack such an implicit adaptation as this work is to our knowledge the first to evaluate them on ImageNet1k. To address this, Beyer et al. (2020) introduced a novel set of validation labels that better capture the multi-label nature, where a prediction is deemed correct if it matches one of the categories present in the image. We observe further very significant improvements of $\approx 7\%$ when employing ImageNet ReaL.

Overall, these results underline that a bad inductive bias as exhibited by an MLP can indeed be overcome if subjected to enough scale. For theory, the results are double-edged; while MLPs prove to be a good proxy to understand transfer learning, data augmentation proves to be a crucial component. Also test-time augmentations significantly boost performance. Both these components on the other hand remain rather understudied in theoretical works.

**Large batch-sizes.** We further make the counter-intuitive observation that training with larger batch sizes significantly boosts performance both up- and downstream. In Fig. 4 we plot pre-training batch size against resulting linear downstream accuracy on CIFAR100 for different number of pre-training epochs. We observe that across all training times, using a larger batch size leads to significantly better performance. Moreover, we want to highlight that such a plot is even favoring small batch-sizes since those models perform more gradient updates for a fixed number of epochs. This effect is in stark contrast to convolutional architectures where entire lines of works have focused on preserving the performance of the small batch-size regime for larger ones (Goyal et al., 2017; You et al., 2017; Hoffer et al., 2017; Keskar et al., 2017). Training with large batch-sizes without degradation is of high interest as it can lead to potentially more efficient training pipelines since computation can be sharded among more devices. This observation about optimal batch-sizes is in-line with similar recent conclusions in Transformers (Kaplan et al., 2020; Touvron et al., 2023).

**Role of augmentations.** Data augmentation is very pronounced for MLPs, providing indirect inductive bias to the model. Remarkably, a model pre-trained on 12 million examples without data augmentation shows inferior performance on CIFAR10 compared to a network trained from scratch

with augmentations turned on. This emphasizes that augmentations go beyond merely leading to a bigger dataset but provide the model with useful invariances. We investigate the learnt weights in Appendix C, showing very evidently, that more localized features are learnt if data augmentation is employed. The power of augmentations has already been demonstrated through the advent of self-supervised learning (Grill et al., 2020; Caron et al., 2021; Chen et al., 2020). Even when training on purely random labels, it still provides powerful learning signals (Anagnostidis et al., 2023).

## 4.3 Scaling Laws

One of the key mysteries in deep learning is that networks tend to improve in terms of generalization when compute, in the form of parameter count and dataset size, is scaled up. Recently it has been observed in several works that the benefits of scale are highly predictable, i.e. generalization performance exhibits a power-law structure when plotted against compute measured in FLOPS (Rosenfeld et al., 2020; Hestness et al., 2017, 2019; Kaplan et al., 2020; Zhai et al., 2022). The functional form has recently been further refined (Caballero et al., 2023). The predictable nature of test performance has even been leveraged to estimate the optimal model before training (Hoffmann et al., 2022; OpenAI, 2023). In order to understand this important characteristic of deep learning theoretically, it is important to analyze whether MLPs exhibit similar properties.

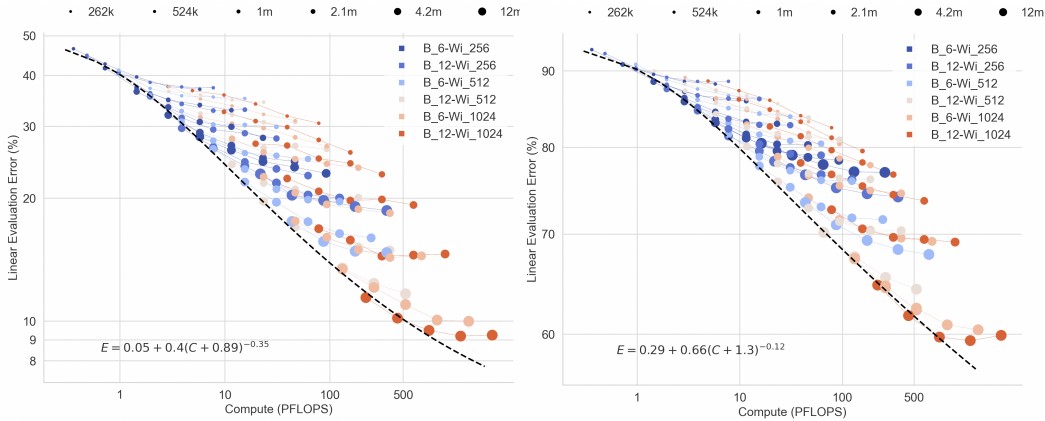

Figure 5: Test error (in %) on CIFAR10 (left) and ImageNet1k (right) when linearly transferred as a function of PFLOPS, measured according to Eq.(4), on a log-log scale.

**Compute.** Following OpenAI (2018) we define the computational cost $C$ incurred from training a model $f$ on $N$ examples for $T$ epochs as

$$C = \text{FLOP}(f) \times 3 \times N \times T, \tag{4}$$

where $\text{FLOP}(f)$ denotes the number of FLOPs needed to complete the forward pass of $f$ for a single example. We note that the number of parameters $P$ present in $f$ enters this equation implicitly in the form of $\text{FLOP}(f) \propto P$. Observe that a given level of compute can be achieved in different ways, i.e. using more parameters $P$, training on more examples $N$, or training for a longer time $T$. When allocating a given level of compute optimally, it is observed that for convolutional and transformer-based architectures, the test error $E(C)$ as a function of compute behaves as a power-law

$$E(C) = a(b + C)^{-\alpha} + E_\infty, \tag{5}$$

where $a, b, E_\infty \in \mathbb{R}_+$ and $\alpha > 0$ is the scaling coefficient determining the rate of decay. $E_\infty$ denotes the irreducible error, i.e. even if infinite compute were employed, the performance remains imperfect. The test error can be measured upstream (i.e. on the pre-training task) or downstream when fine-tuning on a different task. We investigate various pre-training schemes with different number of examples, parameter counts and training times. We subsample ImageNet21k proportionally across classes and pre-train variously sized inverted bottleneck MLPs. We summarize the configurations in Table 3. We then measure test error on the downstream task of CIFAR100 in Fig. 1 as well as CIFAR10 and ImageNet1k in Fig. 5 by linearly transferring the learnt features (without test-time augmentations). The plotting style is inspired by Zhai et al. (2022). Each point in the curve is the downstream performance of an MLP, where the color of the point indicates the model type (blue denotes smaller

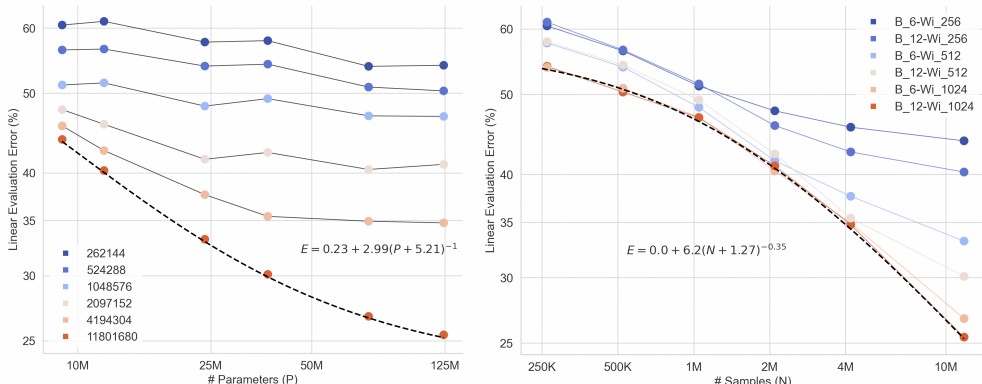

Figure 6: Power law in linear evaluation error on CIFAR100 (in %) when either bottlenecked by the number of parameters (left) or the number of examples (right), on a log-log scale. The dotted line visualizes the fitted functional form.

and red larger models) and the size of the point indicates the number of pre-training examples. Points connected by a line indicates longer training times where $T \in \{50, 100, 200, 400, 800\}$ is measured in epochs. In all experiments, we employ data augmentation for pre-training. We observe that the compute-optimal performance of MLPs strongly exhibits characteristics of a power-law with coefficients $\alpha \in \{0.12, 0.25, 0.35\}$. This is very encouraging for future theoretical work, showing that MLPs indeed mirror the scaling behaviour of modern models. We provide the dual results for the standard MLPs in Appendix B.2, noting that they exhibit essentially the same scaling behaviour, albeit with a slightly weaker slope and intercept.

We further study how performance $E$ evolves when compute is either bottlenecked by the number of parameters $P$ or the dataset size $N$. We visualize the resulting scaling laws in Fig. 6. We find a very steep decay rate in terms of parameters $P$ where roughly $\alpha_P \approx 1$, whereas for dataset size $N$ we identify a significantly slower rate of $\alpha_N \approx 0.35$. This shows that the performance of MLPs is significantly more limited by the dataset size, which is in-line with the fact that MLPs exhibit a bad inductive bias. We investigate the role of dataset size and parameters more in the next paragraph.

**Parameters or examples.** Given a fixed level of compute $C$, what is the optimal way to allocate it to parameter count $P$ and number of examples $N$? To be more comparable to previous work, we assume a fixed training time $T = 50$. To answer this question, we follow the approach outlined in Hoffmann et al. (2022) and plot the optimal compute models identified in Fig. 1 both against model size $P$ and number of examples $N$ and visualize the results in Fig. 7. We empirically observe that the optimal parameter count $P^*(C)$ and dataset size $N^*(C)$ as a function of compute $C$ exhibit power-law behaviour of the approximate form

$$P^*(C) \propto C^{0.35} \qquad N^*(C) \propto C^{0.65}$$

While for transformers, the number of examples (or tokens) $N$ and parameters $P$ are scaled equally (Hoffmann et al., 2022) (i.e. $\alpha_P \approx \alpha_N \approx 0.5$), in contrast we observe that the optimal strategy for MLPs invests significantly more compute into dataset size $N$. This is further evidence for the weaker inductive bias present in MLPs, which needs more examples in order to be compensated for.

## 4.4 Computational Feasibility

We believe that a further exciting feature of our study is its computational feasibility, while at the same time preserving the main characteristics of large-scale pre-training. All of our experiments were conducted on a single NVIDIA RTX A5000 GPU with 24GB of memory. In conjunction with the strongly optimized FFCV dataloading framework (Leclerc et al., 2023) and the inherent efficiency of MLPs, we are able to perform very rapid training. For instance we complete a single epoch on ImageNet21k with the *B-12/Wi-1024* architecture, equipped with 124 million parameters, in only roughly 450 seconds, while the smaller variant *B-6/Wi-1024* at a parameter count of 74 million requires roughly 250 seconds on the specified hardware. Low memory requirements allow us to train with a batch-size of 16384 without having to shard computation among multiple GPUs. We

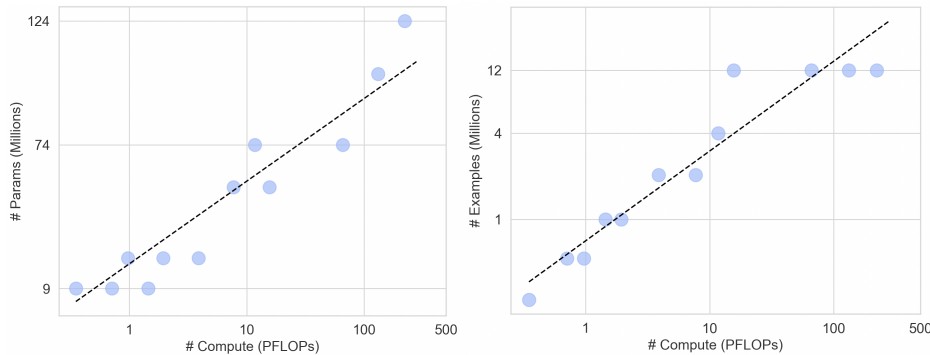

Figure 7: Optimal model size (left) and number of examples (right) for a given level of compute for linear evaluation on CIFAR100, on a log-log scale.

compare the computational efficiency of MLPs with contemporary networks of similar size such as *ResNet-152*, *ViT-B/4* and *ViT-B/8* in Appendix A.5.

## 5 Related Works

There are some prior works that investigate MLPs on vision tasks. Lin et al. (2015) study the performance of MLPs on small scale datasets such as CIFAR10. They observe similar improvements when using inverted bottleneck layers but do not study larger-scale setups, transfer-learning nor do they discuss the implications for theoretical works. The bottleneck structure used in this work has also been investigated theoretically (Parhi and Nowak, 2021; Shenouda et al., 2023; Parkinson et al., 2023), further highlighting that such an architecture exhibits desirable properties. Urban et al. (2017) study to what degree convolutions are necessary for good performance and conclude that even with distillation techniques it remains very difficult to train performant MLPs on CIFAR10. Other approaches have focused on sparsifying fully-connected layers through evolutionary training (Mocanu et al., 2018; Fernando et al., 2016), aiming to learn a good inductive bias from scratch. Similarly, Neyshabur (2020) study how the inductive bias of MLPs can be improved by systematically sparsifying them with a LASSO-type algorithm, making them more convolution-like. d'Ascoli et al. (2019) on the other hand first train a convolutional network for a certain duration and then subsequently continue training the network as an MLP (by using the correspondence between CNNs and MLPs highlighted in Sec. 2). They show that good performance can be reached if the network was trained long enough as a CNN. In contrast to these works, our goal is not to enhance the inherent inductive bias of MLPs but study whether it can be overcome with enough scale.

The advent of the *MLP-Mixer* (Tolstikhin et al., 2021) has led to a series of follow-up work, similarly using MLPs as a patch processor and token mixer (Touvron et al., 2021; Chen et al., 2022; Lian et al., 2022; Guo et al., 2021; Liu et al., 2021). Again, we remark that these architectures all possess significantly more inductive bias.

Finally, we would like to remark that MLPs are successfully used in other areas such as novel view synthesis (e.g. NeRF (Mildenhall et al., 2021)).

## 6 Discussion

In this work, we have explored the limits of the multi-layer perceptron as an architecture for vision tasks. Our study reveals that (1) lack of inductive bias can be compensated by scale and (2) MLPs constitute a (largely) accurate proxy for modern architectures, further cementing their role as the main theoretical object of study. The role of data augmentation and the implicit bias of SGD however strongly differ for MLPs in the setting considered in this work and theoretical works should take this into account. Large-scale pre-training of MLPs proves to be very efficient, enabling researchers with less access to computational resources to study this very exciting line of work. While lack of inductive bias does not prevent MLPs from reaching impressive performance, it leads to an interesting shift in compute-optimality towards more training examples. Subjecting MLPs to even larger amounts of compute similar to Zhai et al. (2022), especially in the form of more training examples, remains as very interesting future work.

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

# Appendix

## A  Experimental Details

### A.1  Resources

For all experiments we rely on NVIDIA RTX A5000 GPU with 24GB of memory. Every experiment can be performed on a single GPU. We leverage the FFCV dataloader framework since the transfer time of the data to the GPU becomes the bottleneck in terms of training time in case of MLPs. All of our experiments were performed in PyTorch (Paszke et al., 2019).

### A.2  Additional Ablations

**Ablations.**    We provide some more ablations in Fig. 8. More specifically, for a (approximate) fixed budget of compute, we investigate different architecture and optimization choices, when pretraining on ImageNet1k and performing linear probing on CIFAR100.

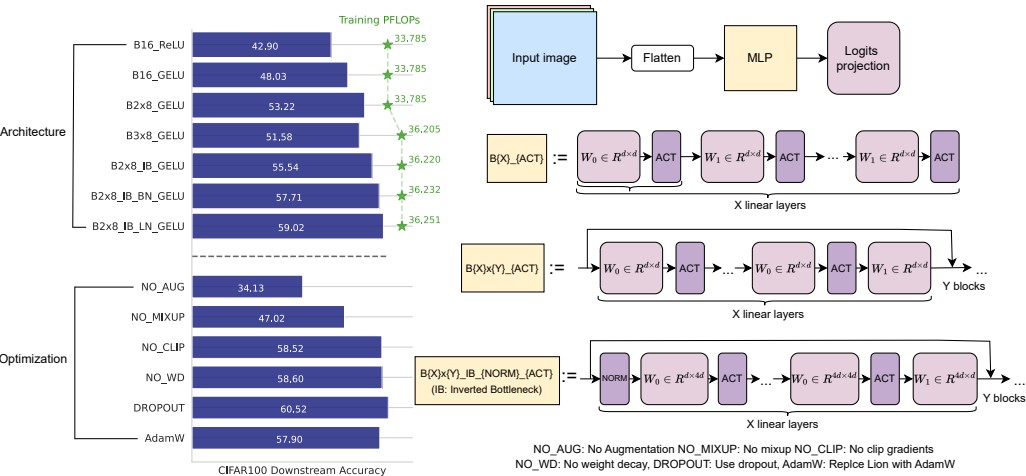

Figure 8: Ablations on different architectures and optimizations choices when training on ImageNet. Numbers indicate linear probing Top-1 accuracies on CIFAR100.

**Normalization.**    We investigate the importance of the normalization method (LayerNorm vs Batch-Norm) in more detail in Table 4. We pre-train two B-MLPs on ImageNet21k with layer normalization and batch normalization and compare the fine-tuning performance on various tasks. We find that the techniques perform similarly, which layer normalisation having a slight edge.

|  | CIFAR-10 | CIFAR-100 | TINYIMAGENET | IMAGENET |
|---|---|---|---|---|
| LAYERNORM | 90.0 | 74.6 | 59.6 | 36.2 |
| BATCHNORM | 89.4 | 73.8 | 57.7 | 35.9 |

Table 4: Pretraining a B-6/Wi-1024 B-MLP with BatchNorm and LayerNorm on ImageNet21k and subsequently fine-tuning.

**Label smoothing.**    We further ablate the influence of label smoothing on the downstream performance. We pre-train B-MLPs with varying amounts of label smoothing ($\alpha \in \{0.0, 0.1, 0.3\}$) and evaluate the resulting down-stream fine-tuning performance. We report the results in Table 5. While label smoothing does provide some boost in performance, the gains are very modest. Label smoothing is thus helpful but not essential for training MLPs.

|  | CIFAR10 | CIFAR100 | TINYIMAGENET | IMAGENET |
|---|---|---|---|---|
| $\alpha = 0.3$ | 90.0 | 74.6 | 59.6 | 36.2 |
| $\alpha = 0.1$ | 89.5 | 73.7 | 58.2 | 36.0 |
| $\alpha = 0.0$ | 89.2 | 72.2 | 57.1 | 35.7 |

Table 5: Pretraining a B-6/Wi-1024 B-MLP with different amounts of label smoothing on ImageNet21k and subsequently fine-tuning.

**Architecture.** We make the following observations/recommendations to boost the model's performance, in line with results reported in the literature (Liu et al., 2022); (1) replacing ReLUs and GELUs boosts results significantly, (2) adding skip connections every two layers helps with optimization, especially for deeper networks. (3) Using an inverted bottleneck increases performance even more. (4) Using a normalization layer in the PRE-LN configuration helps with optimization and (4) layer normalization leads to significantly better results compared to batch normalization, while also being more stable during training.

**Optimization.** As discussed in the main text, augmentations are crucial, and disabling them can have a detrimental effect. We also found that clipping gradients, using weight decay and dropout have a small positive effect on downstream performance. Finally, replacing LION (Chen et al., 2023) with Adam(W), leads to a decrease in performance.

## A.3  Linear Probing

We showcase the transferability of our MLPs by training a linear classifier on top of the frozen features. For training the linear layer, we use the LION optimizer with a learning rate of $\eta = 0.00001$ for 50 epochs. We display the results in Table 6. We observe very strong down-stream performance

|  | CIFAR10 | CIFAR100 | STL10 | TINYIMAGENET | IMAGENET |
|---|---|---|---|---|---|
| *B-6/Wi-1024* | 65.1 | 41.3 | 53.4 | 45.6 | 13.0 |
| *B-6/Wi-1024* + DA | 87.8 | 73.2 | 85.2 | 61.3 | 39.2 |
| *B-12/Wi-1024* + DA | 90.6 | 74.5 | 88.3 | 68.5 | 40.7 |

Table 6: Linear probing Top-1 accuracies when pretraining on ImageNet21k.

even in this more limited setting, highlighting how transferable the features learnt by MLPs are.

## A.4  Scaling Laws

**Implementation Details.** For the scaling law plots, we trained all the models with a batch-size 16384 and the LION optimizer with a learning rate $\eta = 0.00001$ and weight decay of strength 0.001. We further use label smoothing of strength 0.3. We again use augmentations in the form of random flips and crops as well as MixUp with strength 0.8. We rely on the *curvefit* function from the *SciPy* library (Virtanen et al., 2020) to fit powerlaws of the form $E(C) = a(b + C)^{-\alpha} + E_\infty$.

## A.5  Computational Efficiency

We highlight the fact that although MLPs require a lot of training data, inference is extremely efficient from a computational perspective. To illustrate this, we embark on the following comparison; we study inference on $64 \times 64$ resolution images in an MLP vs other popular vision architectures of similar size and complexity in Table 7. More specifically, we compare against a *ResNet-152*, where we replace the stride in the first convolutional layer and remove the first max-pooling operation to

compensate for the smaller image size. We also compare against a base *ViT* and *Mixer* model, where we extract patches from $4 \times 4$ regions in the original image.

As it quickly becomes eminent, MLPs require significantly less FLOPs to make predictions on individual images, in essence utilizing their parameters a lot more methodically. As a result, latency and throughput are significantly better compared to other candidate architectures. We measure throughput using the optimal batch size on an NVIDIA RTX A5000. We highlight, that our MLPs, in contrast to the other architectures are memory bound, meaning that their throughput is determined by the prefetching bandwidth of our GPU. Hardware advancement and specialized architectures could significantly mitigate this effect. Neglecting memory transfer time by propagating the same input through our network gives a further 6-fold increase in the potential throughput.

| | PARAMETERS | LATENCY (MSEC) | THROUGHPUT (IMAGES/SEC) | FLOPS PER FORWARD PASS |
|---|---|---|---|---|
| *B-12/Wi-768* | 66.89 *M* | 21.2 | 16063 | 66.8 *M* |
| *ResNet-152* | 60.19 *M* | 423 | 506 | 13.07 *G* |
| *ViT-B/4* | 86.06 *M* | 424 | 222 | 23.08 *G* |
| *Mixer-B/4* | 63.82 *M* | 400 | 319 | 19.36 *G* |

Table 7: Various measures assessing the computational efficiency of different architectures.

## B  Results for Standard MLPs

### B.1  Transfer Learning

For completeness we also analyze the transfer performance of standard MLPs when pre-trained on ImageNet21k. We compare a S-MLP of depth $6$ and width $2048$ against a B-MLP of depth $6$ and width $1024$, ensuring that both models roughly have a parameter count of around $\approx 70$ million. We display the results in Table 8. We observe that even the features of a standard MLP (i.e. without residual connections and bottleneck structure) transfer very well on different downstream task. Tthe inverted-bottleneck MLP however still remains superior.

| | CIFAR10 | CIFAR100 | TINYIMAGENET | IMAGENET |
|---|---|---|---|---|
| S-MLP | 87.1 | 68.3 | 52.1 | 30.2 |
| B-MLP | 90.0 | 74.6 | 59.6 | 36.2 |

Table 8: Comparing a S-MLP of width 2048 and depth 6 pre-trained on ImageNet21k, with a B-6/Wi-1024 B-MLP (both models around 70M params) in terms of fine-tuning performance

### B.2  Scaling Laws

We also evaluate the scaling law of standard MLPs by training variously sized models on different subsets of ImageNet21k and subsequently linearly probing the features on CIFAR100. The setting is identical to the one described in 4.3. We observe that also standard MLPs exhibit power-law behaviour. The slope ($0.22$ vs $0.25$) and the intercept ($0.18$ vs $0.16$) are however worse when compared against the inverted-bottleneck MLP.

## C  Weight visualizations

We visualize the first layer weights $\boldsymbol{W}^{(1)} \in \mathbb{R}^{3wh \times m}$ by reshaping them back to $\mathbb{R}^{w \times h \times 3 \times m}$. We then produce a $\mathbb{R}^{w \times h \times m}$ representation by taking the maximal value along the channel dimension.

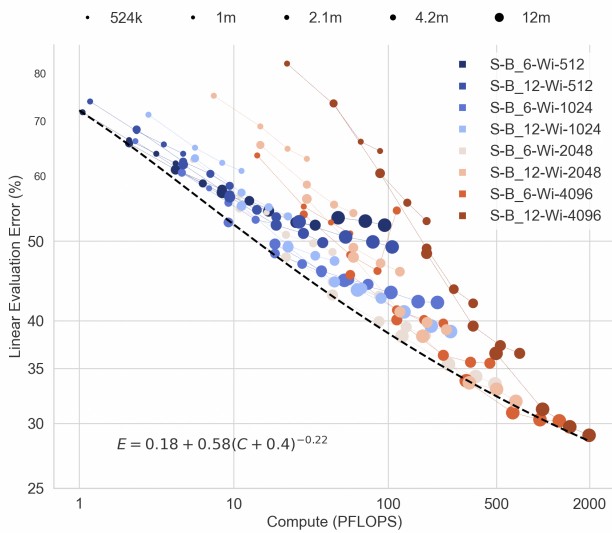

Figure 9: Test error of standard MLPs on CIFAR100 when linearly transferred as a function of PFLOPS, measured according to Eq.(4), on a log-log scale.

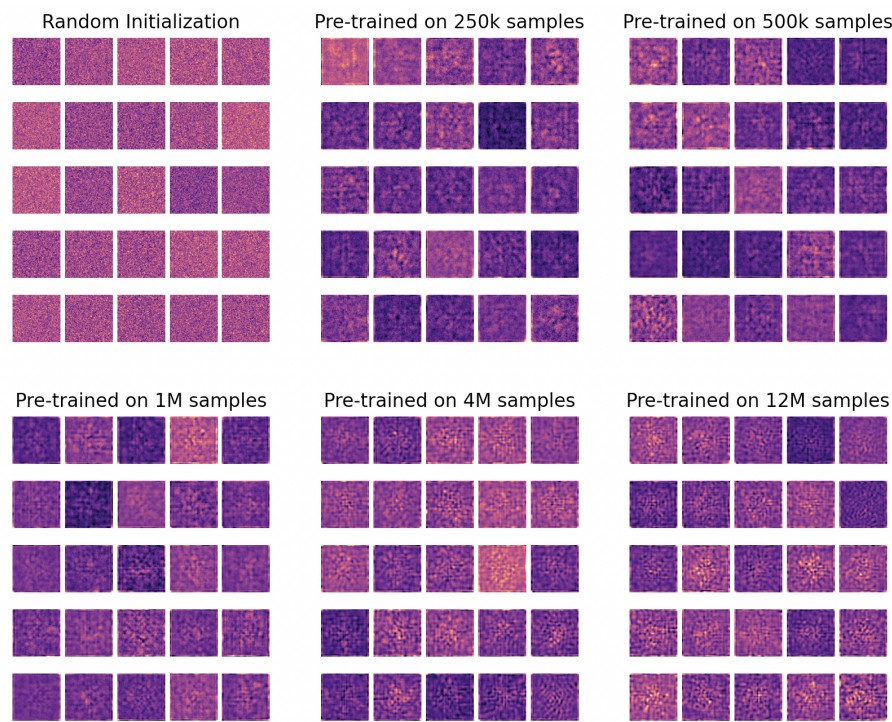

Figure 10: Visualization of the first layer weights for different pre-training dataset sizes.

We display such visualizations of the first $5 \times 5 = 25$ "filters" for different pre-training sizes, also including the weights at random initialization in Fig. 10. All models were trained with data augmentation. We observe that filters increasingly develop structure as we increase the dataset size and become more and more localized. We further compare against models that were pre-trained on the full ImageNet21k, with and without data augmentation in Fig. 11. We observe that even though we provide the model with an abundance of samples, the weights still remain largely structure-less

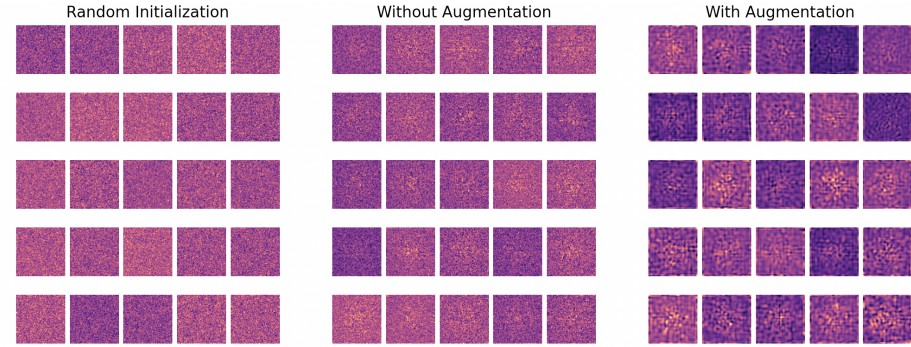

Figure 11: Visualization of the first layer weights for models trained with and without data augmentation.

and have not developped any locality properties. On the other hand, using data augmentation leads to more adapted filters.

## D  Inverted Bottleneck MLP Code

We provide PyTorch-style pseudo-code for the inverted bottleneck MLP to highlight its simplicity.

```python
from torch import nn

class Block(nn.Module):
    def __init__(self, dim, expansion_factor=4, dropout=0.):
        super().__init__()
        self.fn = nn.Sequential(
            nn.Linear(dim, int(expansion_factor * dim)),
            nn.GELU(),
            nn.Dropout(dropout),
            nn.Linear(int(expansion_factor * dim), dim),
            nn.Dropout(dropout)
        )
        self.ln = nn.LayerNorm(dim)

    def forward(self, x):
        return x + self.fn(self.ln(x))

def MLP(image_size, channels, dim, depth, num_classes,
        expansion_factor=4, dropout=0.):
    return nn.Sequential(
        nn.Flatten(start_dim=1, end_dim=-1),
        nn.Linear(image_size * image_size * channels, dim),
        *[Block(dim, expansion_factor, dropout) for _ in range(depth)
    ],
        nn.Linear(dim, num_classes)
    )
```

