# OpenReview forum: "Scaling MLPs: A Tale of Inductive Bias"
_NeurIPS.cc/2023/Conference — NeurIPS 2023 poster_

### Official Review · Reviewer_EAzm · 2023-06-28

**Soundness:** 2 fair
**Presentation:** 4 excellent
**Contribution:** 2 fair
**Rating:** 4
**Confidence:** 4

**Summary:**

This paper argues that MLP networks can perform well on challenging tasks when they are scaled up. Thus, they are able to compensate for their lack of inherent inductive bias when trained on a large amount of data with many model parameters. The authors show through experiments on popular image classification datasets that indeed, MLPs can perform surprisingly well when scaled up. Along the way, the authors make a number of other observations regarding the role of MLP architecture, data augmentations and scaling laws.

**Strengths:**

**Originality**

The question the authors aim to answer is quite interesting. As far as I can tell, few empirical studies have approached the question of whether MLPs can be made to perform competitively on standard benchmarks under large-scale data and model size (although studies have trained MLPs on standard benchmarks for other reasons).

**Quality**

The experiments are extensive and thorough. The authors train on ImageNet21k and evaluate on standard image classification benchmarks; in my view, these are exactly the most valuable experiments to run to investigate the question the authors aim to answer.

**Clarity**

The paper is quite well-written. The key motivation of the paper and background are made very clear, and figures are well-illustrated. As a minor suggestion, moving Table 1 and 2 closer to the experiment section would be helpful.

**Significance**

Overall, if the paper can demonstrate its claims properly, the paper would be quite significant to the field. Showing that the inductive bias of CNNs or Transformers can be compensated with scale will perhaps focus the attention of the field toward building inductive biases that can't as easily be compensated by scale, as well as validating (or invalidating) the idea of applying theory done on MLPs to other models.


**Weaknesses:**

My main concern with the paper is with the interpretation of the experimental results; I'm concerned that the results are not enough to support the authors' hypothesis 3.

First, the authors only conduct experiments on image classification tasks. If the authors wished to make a more general claim about inductive bias in MLPs vs other models, it would be best to conduct experiments on other domains. Alternatively, the authors may wish to restrict their claims to the image domain.

Second, the empirical results on ImageNet do not appear quite strong. In Table 4, the authors find a maximum ImageNet top-1 accuracy of 41%; this does not seem very competitive. If the authors wanted to show that MLPs can indeed circumvent their lack of inductive bias, I would expect to see a result showing near state-of-the-art (or at least competitive results) on ImageNet. I may be missing some key empirical results though.

Third, the authors find that data augmentation is critical to the performance of MLPs. Data augmentation, as the authors note, allows the model to learn invariances that would otherwise be built into a model like a CNN. Thus, data augmentation is a powerful inductive bias. However, this fact seems to undermine the authors' claim that scale alone is sufficient to compensate for lack of inductive bias. The authors may wish to reformulate their claim to suggest that architectural inductive biases can be compensated by a combination of scale and other inductive biases.

Similarly, the authors find that using a specific inverted bottleneck MLP architecture is important for performance. The authors seem to view this as a minor addition to the MLP, but given the large performance improvement, this addition is actually quite significant. Indeed, in my view, architectural changes that are simple but lead to large performance gains are exactly those that are most valuable to study!

Also, as a more minor point, the authors claim that MLPs are "completely free of any inductive bias." This seems to be too strong of a claim: they are often one of the simplest models to implement, but this doesn't mean they are free of any inductive bias. The claim quoted by the authors from the *Principles of Deep Learning Theory* book seems to be a more fair statement to me.

**Questions:**

Can performance on ImageNet be improved to near state-of-the-art for MLPs?

What is the relative significance of scale vs. data augmentation/model enhancements for MLPs?

**Limitations:**

It would be good if the authors could mention some of the limitations listed above in the discussion section.

No potential negative societal impacts.

---

> ### Author Rebuttal · Authors · 2023-08-08
>
> We thank the reviewer for the feedback and the interesting points raised. We will address them in the following.
>
> **Image classification:** We agree with the reviewer and we will highlight that we focus on image classification. We largely do this since inductive bias is clearer and more intuitively defined in the image domain, compared to other domains as for instance natural language processing.
>
> **ImageNet performance:** We agree with the reviewer that ImageNet performance is not competitive with standard baselines. We suspect several reasons for this:
>
> 1. Due to computational constraints, we down-sized images to 64 x 64, making the fine-grained classification required for ImageNet more difficult.
> 2. We also want to highlight that our work is the first with the aim of making MLPs competitive. Other architectures like convolutions have seen a decade of development from the entire ML community, leading to very specialized recipes (optimizers, learning rate schedules, normalization methods etc.). While AlexNet still outperforms our MLPs (achieving around 55% https://pytorch.org/hub/pytorch_vision_alexnet/ ), this gap is significantly smaller and we believe that similar progress could be made for MLPs with time.
>
> Finally, we also want to stress that MLPs achieve very strong downstream performance on every other task, thus indeed highlighting that lack of inductive bias can be overcome. We hypothesize that training on a larger dataset like (the private) JFT-3B with 3 billion images would significantly boost the performance on ImageNet.
>
> **Data augmentation:** This is a good question! We agree with the reviewer that data augmentation allows us to implicitly encourage networks to learn invariances relevant to the task, thus adding some level of inductive bias. We believe however that the regularizing nature of augmentations is more crucial for the case of MLPs, since training accuracy remains far from perfect even after long periods of training (see Table 5 in the rebuttal pdf). The MLPs hence never really learn the invariance in the data but at the same time also don’t overfit, which helps performance. We do however acknowledge that training with data augmentation does add inductive bias implicitly. In this work we largely focus on the inductive bias of the model architecture, as the reviewer correctly highlights. Moreover, even if data augmentation is employed, scale is still needed to reach decent performance and overcome the lack of inductive bias in the model. We will rephrase the text to make this clearer.
>
> **Bottleneck MLP:** This is a great question. We managed to significantly improve performance of V-MLPs (the training protocol was sub-optimal before). We observe that they become more competitive with B-MLPs when training from scratch, thus downplaying the importance of the bottleneck layer (see Table 1 in attached pdf). We also pre-trained a vanilla MLP of width 2048 and depth 6 on ImageNet21k and compare it with a B-6/Wi-1024 BottleneckMLP in terms of fine-tuning performance (see Table 3 in attached pdf). We chose different widths to make the parameter count comparable (both around 70M). We observe that BottleneckMLPs generalize better in accordance with previous literature but vanilla MLPs also achieve strong performance, further showing that this design choice helps but is not crucial. We have also evaluated the scaling laws for vanilla MLPs on CIFAR100 (see Figure 1 on the left) and find that it is very similar with a slightly worse slope (0.22 vs 0.25), as expected. We think this further strengthens our claims regarding inductive bias and also makes our findings more directly compatible with theory settings.
>
> **Inductive bias of MLP:** We agree with the reviewer and we regret our overly strong wording regarding the inductive bias of MLPs. Comparatively to CNNs and ViT, MLPs have a very weak inductive bias, they are for instance not vulnerable to permutations of pixels. However, they still encourage a hierarchical order of features, which does serve as an inductive bias (albeit not specialized towards vision in particular). We will rephrase the text accordingly.
>
>
> We again thank the reviewer for the interesting points raised, leading to a refinement of our claims and statements, as well as disentangling the roles of different components more carefully.

---

> > ### Comment · Reviewer_EAzm · 2023-08-11
> >
> > Thank you for your response.
> >
> > I think the additional experiments are valuable, particularly those showing that V-MLPs are quite effective and those showing the performance on ImageNet1k at varying resolutions. Both of these strengthen the claim that vanilla MLPs can be effective on challenging image classifications with sufficient scale. These alleviate some of my experimental concerns. I've increased my rating correspondingly.
> >
> > Nevertheless, I'm not fully convinced that just because MLPs achieve strong performance on easier tasks, they will also achieve strong performance on ImageNet given enough scale. Moreover, theoretically, it is not surprising that sufficiently large MLP with sufficient amounts of data can learn ImageNet (or any dataset); it is the empirical demonstration of this fact that is most interesting, and which I think would be the most valuable addition to this paper.
> >
> > Also, I'd like to highlight that refining the claims made by the authors (as they have noted in their response) will be critical in the revision so that the claims properly match what can be justified by the experiments and prior work.

---

> > > ### Author Response · Authors · 2023-08-14
> > > **Response**
> > >
> > > We really appreciate that the reviewer took the time to read and discuss our rebuttal, and thank the reviewer for increasing the score.
> > >
> > > We acknowledge that for the complex task of ImageNet, the test accuracies of our best model alone do not fully indicate that MLPs will overcome the lack of inductive bias on this task. We have however also evaluated the scaling law (i.e. how test performance of a linear probe varies as a function of compute) in Figure 10 in Appendix A.5, which predicts that MLPs will reach $\approx 71$% test accuracy when enough compute is used. This is very competitive with other models, especially given that:
> > >
> > > 1. This is still using the very limiting resolution $64 \times 64$ vs. the more standard $224 \times 224$.
> > > 2. This applies for the less powerful linear probes, not the usual full fine-tuning of the model.
> > > 3. More “empirical tricks” as developed for other architectures like CNNs would further improve this scaling.
> > >
> > > We acknowledge that scaling laws are not guaranteed to extrapolate accurately but previous literature has shown that they are surprisingly robust and can even be used for architecture search [1]. We thus believe that our experiments do provide further insights into how MLPs scale, but we unfortunately lack the resources to demonstrate it empirically at such large scales.
> > >
> > > Finally, we also would like to highlight that the aim of overcoming lack of inductive bias is only one facet of our work. We believe that providing the theoretical community with data points as to how MLPs perform and behave in modern settings (no matter whether competitively or not on a given dataset) is still an important contribution. We show that MLPs do largely behave like their more complicated modern counterparts (with a few important exceptions), which cements their role as a theoretical model. We will also publish the pre-trained checkpoints so that researchers can investigate the properties of their theoretical model more closely and gain more insights into its inner workings. This might inspire more theoretical works guided by our empirical results.
> > > \
> > > \
> > > [1] GPT-4 Technical Report, OpenAI, 2023

---

### Official Review · Reviewer_RAGY · 2023-07-06

**Soundness:** 4 excellent
**Presentation:** 4 excellent
**Contribution:** 4 excellent
**Rating:** 8
**Confidence:** 5

**Summary:**

This paper presents a comprehensive study on the scaling capabilities of MLPs, challenging the conventional belief that this architecture is limited in its performance on vision tasks. The authors employ recent advancements in deep learning architectures to modernize MLPs while preserving their core characteristics. The experimental results obtained in this study are remarkable, indicating that tasks previously deemed impossible for MLPs can now be accomplished.

To achieve these results, the authors incorporate several crucial components of modern deep learning architectures into MLPs. They introduce normalization layers, skip connections, bottlenecks, and a large batch size, along with a modern optimizer. These additions ensure stable training, thus allowing to train on large datasets. Additionally, the authors demonstrate that despite the absence of inherent inductive bias in MLPs, data augmentation and sufficient data volume effectively compensate for this limitation. Consequently, scaled MLPs perform competitively with architectures possessing stronger inductive biases, such as CNNs or ViTs, on several datasets, while being much more computationally efficient at a fixed resolution.

Finally, the authors assess the scaling properties of MLPs providing accurate estimates of their scaling laws in terms of data budget and compute budgets. Unlike other architectures, in scaling MLPs larger datasets are more important than compute budget. The findings open up new possibilities for utilizing MLPs in vision tasks and highlight the importance of reconsidering their scaling capabilities in the broader context of deep learning research.

**Strengths:**

#### 1. **Addressing a clear research gap in the literature**:
This paper successfully completes the research program started with the ViT paper and continued with the MLP-Mixer and ConvMixer that challenges common assumptions regarding the benefits of embedding inductive biases in architectures. By answering the question of whether inductive bias can be fully circumvented with more data affirmatively, this paper fills a significant void in the literature. Although the results might have been expected by many, the clear empirical validation of this hypothesis makes this paper an important contribution.

#### 2. **Thorough experiments with solid results**:
The execution of the experiments and the manner in which they are conducted are exemplary. The paper provides a comprehensive ablation analysis across various settings, thoroughly examining and validating each claim. The results obtained are both convincing and transparent. Notably, the authors openly discuss the relatively poorer performance on ImageNet and offer insightful explanations for this observation, enhancing the credibility of their findings.

#### 3. **Outstanding presentation**:
The paper exhibits outstanding writing quality, making it a pleasure to read. The clarity of the language and the structure of the paper contribute to its overall excellence. The authors effectively communicate their ideas, methods, and results, ensuring that readers can easily comprehend and appreciate the significance of their work.

**Weaknesses:**

#### 1. **Missing discussion regarding the importance of processing images in patches**
In my opinion, the paper is missing an important discussion regarding the critical factor that makes MLP-mixers superior to the presented full MLP architecture, i.e., the processing of images in patches. Although I understand a very in-depth empirical analysis of these differences might be outside of the scope of the paper, it is important to fully acknowledge the importance of this key design choice. In particular, I believe it is important to also reference the ["Patches are all you need?"](https://openreview.net/forum?id=rAnB7JSMXL) paper which also tried to fill a research gap in this sense prior to this work.

#### 2. **Missing discussion on the effect of image resolution on the results**
It seems that an important design decision in this paper has been the downscaling of all datasets to a 64x64 resolution. I believe this decision deserves further motivation in the text as well as some reflection on the fact that MLPs cannot be applied to inputs of different sizes.

**Questions:**

I would encourage the authors to address the stated weaknesses in the rebuttal and in any future revision of the manuscript. I honestly believe doing so will make the paper better.

**Limitations:**

Overall, I believe the paper has addressed well most of its limitations and that the authors have been open and honest in describing the shortcomings of their results. However, as mentioned in Weakness 2, I believe it is important that they also discuss how resolution and different image sizes matter in their results.

---

> ### Author Rebuttal · Authors · 2023-08-08
>
> We thank the reviewer for the thorough assessment of our work and the very positive feedback. We will address the questions in the following.
>
> **Role of patch size:** We completely agree with the reviewer and also believe that breaking the images into patches plays a fundamental role for all these architectures, as it adds a strong inductive bias. Unfortunately, investigating the role of patchifying is out of the scope of our current work but makes for very exciting future work. Especially investigating the role of the patch-size in ViTs and MLPMixers is very interesting as it modulates the degree of inductive bias in the architecture. We will definitely expand the corresponding discussion and add the very relevant proposed related work. Another interesting work in that regard is [1], which shows that even simple spatial pooling for token-mixing suffices for good performance. The spatial pooling of course adds back a lot of inductive bias, but again relying on patches seems crucial for good performance.
>
> **Resolution:** We completely agree with the reviewer and we will add a paragraph to discuss the role of the resolution. In our work, this decision was largely made due to computational constraints, as a high resolution leads to a very large first layer. We will highlight that MLPs indeed cannot be used directly for different resolutions, thank you for pointing this out! Inspired by this comment, we adapted the resizing technique from FlexiViT [2] in order to enable the MLP to handle different resolutions. We take a B-12/Wi-1024 B-MLP which is pre-trained on ImageNet21k at a resolution of $64 \times 64$ and fine-tune it on ImageNet1k at a resolution of $64 \times 64$. We then evaluate test performance on ImageNet1k for resolutions $16,32,48,64,96,128$ while resizing the embedding layer as outlined in [2] in Section 3.4 (without any further training). We show the results in Figure 1, on the right. We observe that test performance remains stable for all resolutions and slowly starts deteriorating only for the smallest ones. We will include these results in the discussion.
>
>
> [1] MetaFormer is Actually All You Need for Vision, Yu et al.
>
> [2] FlexiViT: One Model for All Patch Sizes, Beyer et al.

---

> > ### Comment · Reviewer_RAGY · 2023-08-15
> >
> > I thank the authors for engaging with my comments and the ones from the other reviewers. In my opinion, the new results strengthen the message of this very good paper.
> >
> > Regarding the main concern of [Reviewer EAzm](https://openreview.net/forum?id=R45A8eKcax&noteId=lR9CmYB1Ev) that there is not enough evidence to suggest that properly scaled MLPs will achieve near state-of-the-art performance on ImageNet, I want to emphasize that already the current provided improvement on ImageNet is strong, and that the results on smaller datasets are very remarkable. In my opinion the fact that this paper keeps the door open to achieving great results on ImageNet by extrapolating their empirical scaling laws is enough reason to accept it. In particular, the authors do not overclaim their results and never say they can achieve such performance already. The paper simply provides a thorough empirical evaluation of scaling techniques with MLPs that nicely complement prior results in the literature. In my opinion, the great execution of the experiments and promising results places this paper clearly above the threshold of acceptance to NeurIPS.

---

### Official Review · Reviewer_bQKr · 2023-07-06

**Soundness:** 2 fair
**Presentation:** 2 fair
**Contribution:** 3 good
**Rating:** 6
**Confidence:** 3

**Summary:**

This paper studies scaling of MLPs on various image classification datasets with and without pretraining and data augmentation.

**Strengths:**

MLP scaling trends are a welcome data point for practitioners and theoreticians alike.

**Weaknesses:**

The paper does not carefully tune the learning rate, weight decay, and other hyperparameters, so the trends shown in this work may not be a Pareto frontier.

**Questions:**

The abstract states "Given the recent narrative less inductive bias is better ... it is natural to explore the limits of this hypothesis" in L4 but this claim is not supported. Rather, a weaker version stated in L52 is investigated in this work, which could be clarified.

The paper states in L7-12 that while deep learning theory literature almost always studies MLPs, there are no empirical datapoints connecting to theory. This sounds like the paper would cite relevant theory works and try to confirm/refute them, but this is not the case. The paper explores scaling of MLPs. At least, the thought in L7-12 does not connect smoothly to the question raised in L12-13. Perhaps, this statement could be softened a bit.

L15 Perhaps, it should be clarified that pretraining on ImageNet21k is used: "... drastically improves with scale [and pretraining ...]"

In Table 3, the parameter count (in the first layer) is resolution dependent, but the resolution is not given.

L194. MLP does indeed become better, but not enough to state that it's competitive with a ResNet.

L231-232. It is common knowledge that FLOP = 2P for MLP

L236. (5) doesn't have the coefficient c. Also, power-law is defined as `a * x ^ power + b` without the affine offset for `x`.

L296. Discussion should be Conclusion.

I don't fully understand why Block(z) = z + ... is called an inverted bottleneck MLP. Residual MLP seems more natural


**Limitations:**

No, the authors didn't write potential societal impact of their work.

---

> ### Author Rebuttal · Authors · 2023-08-08
>
> We thank the reviewer for the helpful feedback on our work. We will address the concerns in the following.
> \
> \
> **Weaknesses:**
>
>
> **Not tuning hyper-parameters:** We apologize for not clearly stating this in the paper but we did tune hyper-parameters employed upstream based on the upstream validation accuracy. We only fully trained models that improve over the current optimal model and otherwise ended runs early on in training. We agree that solely relying on such upstream-tuning might not always be optimal for downstream performance but given our limited computing budget, fully training every hyper-parameter configuration upstream is impossible and we believe that our approach is a reasonable approximation. [1] explicitly explore this question for Vision Transformers on ImageNet21k and find that
>
> *“The results are mixed but generally reflect that the cheaper strategy [upstream validation] works equally well as the more expensive strategy [downstream validation].”*
>
> We also want to highlight that this is a common strategy in many previous works on scaling laws where performing multiple runs for huge models is also impossible [2, 3, 4].
> \
> \
> \
> Given that this was the only concern the reviewer raised, we would be very grateful if the reviewer could further voice avenues for potential improvement of our work or consider raising the score.
> \
> \
> [1] How to train your ViT? Steiner et al., 2021
>
> [2] Scaling laws for large language models, Kaplan et al., 2020
>
> [3] Training Compute-Optimal Large Language Models, Hoffmann et al., 2022
>
> [4] Scaling Laws for Autoregressive Generative Modeling, Henighan et al., 2020
> \
> \
> **Questions:**
>
>
> **Weaker Hypothesis:** Thank you for pointing this out, we will clarify our focus in this work in the abstract.
>
> **DL Theory:** We apologize that we didn’t write this clearly and will improve the text. We do not aim to question the validity of the theoretical works themselves directly. The cited theoretical works focus on strongly simplified settings with the MLP serving as the model of choice (e.g. studying generalization in a synthetic toy task instead of realistic image classification). These simplified settings are still far from real-world settings (e.g. classification on ImageNet) but with time, progress towards that goal can hopefully be made. Our work serves more as a check to ensure that the MLP actually exhibits the interesting characteristics of modern models in the relevant realistic settings. This is the end goal that theoretical works (slowly) try to build up towards. We will make this clearer in the text.
>
> **L15:** Thank you for highlighting this, we will clarify that pre-training on ImageNet21k is used.
>
> **L194:** We will weaken the phrasing in order to avoid confusion.
>
> **Resolution-Dependence:** Thank you for pointing this out, we will clarify the dependence on the resolution as well as the value of the resolution, which is 64 x 64.
>
> **Minor Changes:** Thank you for the improvements on the manuscript, we will incorporate them.

---

> > ### Author Response · Authors · 2023-08-21
> >
> > Dear reviewer **bQKr**,
> >
> > As the discussion phase ends today we will not be able to further clarify potential additional concerns. We would be very grateful if you could respond to our rebuttal and offer us an opportunity to further engage with your concerns and address any additional questions you might have!
> > \
> > \
> > Best,\
> > Authors

---

### Official Review · Reviewer_xaDV · 2023-07-07

**Soundness:** 3 good
**Presentation:** 3 good
**Contribution:** 3 good
**Rating:** 6
**Confidence:** 5

**Summary:**

This paper studies the abilities of Vanilla MLPs and a variant of the MLP with residual connections, LayerNorm, and inverted bottlenecks, performs when scaling in size, compute, batch size, and dataset size. Many learning theory papers study MLP-based settings and claim to motivate their results by deep learning results that are usually run on CNNs or transformers. Somewhat counterintuitively, there are not many clear results on MLPs that showcase all of the relevant behaviors of interest that are studied on more complicated architectures. To justify the theoretical avenues of research further, we would hope that scaling MLPs behaves similarly to scaling CNNs and transformers. Furthermore, one goal of this paper is to understand the claim that the transformer is better due to lack of inductive bias, being a move from a strong inductive bias of CNNs to a weaker inductive bias. Due to the lack of inductive bias, it is often claimed that transformers require further pretraining, but in this regime end up performing better.
The authors study a variety of these claims as they pertain to MLPs and show that their “modern” version of the MLP can, indeed, showcase powerlaw generalization scaling with total compute, as well as improved performance with size of the model and dataset size, as well as the ability to be pretrained and effectively transferred to downstream tasks (all on visual recognition tasks).

EDIT: Thank you for your rebuttal, I think with these changes and clarifications it is a good contribution to NeurIPS and I intend to keep my score and continue to lean towards accept rather than reject.

**Strengths:**

This is a question I myself was excited to see studied. I bid on this paper as I am highly curious to what extent MLPs share the same behaviors as their more modern counterparts. Having seen many learning theory papers that study the MLP, it is a very natural question to know whether they share some of the interesting dynamics that CNNs and transformers show. I think the experiments are largely beginning to answer this question, though I do have many questions and some potential concerns about experiments and the presentation of results in the paper (which I have written in the questions section) and would appreciate more clarity in order to feel very confident in accepting this paper.

I am generally impressed by the results that are shown, especially the transfer learning results, in that an MLP can, in some settings, achieve such accuracies on tasks like CIFAR-10 and CIFAR-100. In some of these settings I am not yet sure the authors have gone deep enough (discussed more below), but I am overall excited about this type of work and want to see more of it. It seems natural to have tried many of these experiments, and yet I can’t recall any paper that has done so.

Clearly, the authors have identified a few key examples of settings in which model scaling is commonly studied, and they have given compelling evidence that some MLPs do indeed follow our existing intuitions. I would love to see a lot more evaluations and discussions, and I think the current results could be a launching pad for more thorough work. As such, I feel that this paper leaves a lot of questions to be answered, but in that sense I think it might still be a worthwhile contribution to the community in its current form, and will help motivate a much-needed bridge between theory and practice, and further inspire more detailed empirical research. However, a revision that carefully addresses the questions I’ve left below would go a long way to convince me that it is a very strong piece of work.

**Weaknesses:**

The authors are motivated by the question of whether MLPs reflect the empirical observations of practical models, but I have some concern of how they qualify practical. These results show some settings of interest that to me seem fairly specific, and try to discuss what is practical or not. I think it is overall a little bit shallow in total experimentation, given it is a purely empirical paper and experiments are all using one GPU I think more can be done. Generally, I am averse to criticizing a paper for not enough experiments and this paper, I feel, has shown sufficient evidence to convince me there is something interesting. But I think that a lot more work has to be done to really show many of the dimensions by which MLPs scale, and how it relates and compares to modern architectures.

I know the comment above may be a little vague, so I list below the various specific questions (in the questions section) that I have upon reading this paper, which I hope the authors will find helpful. I think the questions below subsume anything I could write on weaknesses, and I think the paper would be very strong if it has answers to these questions.

**Questions:**

- Nitpicky: I am not sure I agree that MLP-Mixer has “arguable even less inductive bias” than a Vision Transformer, but this is a very subjective opinion and I could see an argument for it, but given that data is pre-processed into patches and there is mixing between patches and features, it feels like there is quite a similar level of “inductive bias” between the two, and so I am hesitant to accept this claim at face value, but I don’t mind if it is left in the paper either
- The authors make the statement at least twice early on in the paper (e.g. top of page 3) that “MLPs invest their compute significantly more into dataset size compared to model size” and I don’t think I understand what this means, nor did I feel I had much clarity on this statement after reading the experiments. Is it saying that scaling dataset size is more important than scaling model size? If that is the case, I don’t think any of the current plots very clearly justify this point, perhaps you can find a way to make it more clearly
- Why do you use LayerNorm and not BatchNorm? Do you compare to BatchNorm? Does it matter, and if it does how much? How does this relate to normalizations and choices in other models? E.g. does CNN or ViT have less susceptibility to be affected (in how it scales) by normalization layer choices?
- Do you need to use a bottleneck embedding layer? This seems like a pretty significant requirement. Do these scalings not work without a bottleneck embedding? If that is the case I think it is important to be very clear that a specific MLP gives this scaling, but then if you were to take an MLP with wider layers than input dimension (no bottleneck embedding) these results don’t hold. This seems very important to emphasize in your study, as many learning theory papers (that I’ve come across) don’t necessarily distinguish that MLPs need to have a bottleneck embedding layer
- Strong label smoothing also seems like a strong constraint, does this work without it? I think generally label smoothing I can believe is practical but not so aggressive as to use alpha=0.3. If these behaviors are only specific to bottleneck embeddings with strong label smoothing I think that should really change the tenor of the main claims, because you are studying a highly specific setting in which MLPs showcase such scalings, but that an average MLP trained on one-hot encoded labels with wide layers, for instance, does not do this (this seems very important to make clear)
- Perhaps there is a citation or another line of work that gives precedent for aggressive label smoothing like this? You don’t provide any citation or justification of this choice
- A bit related to the last point, you mention combatting overfitting and severe overfitting, but you haven’t reported any training accuracies nor generalization gaps. Throughout Table 1 it is hard for me to totally understand what is overfit vs. underfit when you only report test accuracies.
- Furthermore, you don’t mention whether you use a validation set or any cross-validation when experimenting with different choices, and then report the final error on a held-out test set. Can you clarify your experimental methodology?
- In Table 1 why do you train Vanilla at 100 and then inverted + DA at 1000 and 5000, what happened to Vanilla at 1000 and 5000? Plus Vanilla + DA, Inverted, and ResNet18 + DA don’t report how many epochs it is trained for. Maybe reporting train accuracies and/or generalization gap will help.
- It seems like it would be useful, when comparing different settings, to stop model training at the same train accuracy to control for other factors, is this something you do? Or how do you justify the comparisons you are making? It is hard for me to get a total sense of all of the dynamics at play here purely from test accuracies.
- One of your claims is that larger batches significantly boosts both up and downstream performance, and Figure 3 is plotting Error vs. batch size showing this but the line in page 6 says “we plot pre-training batch size against resulting linear downstream accuracy…” which I just found a bit confusing because when I read that and looked at the downward trend in Figure 3 my immediate thought was that accuracy was going down, but then I saw you had error on the y-axis and not accuracy, so maybe you can reword this line
- In the “Role of augmentations” section you mention that data augmentation gives indirect inductive bias, but you don’t really qualify what this means nor give any citation or justification of this. I guess it reads just like a statement you can make because it’s vague, but I don’t know what exactly you are trying to say to me and why it is correct. Perhaps you can give some intuition on what is left implicit here

---

> ### Author Rebuttal · Authors · 2023-08-08
>
> We thank the reviewer for the very extensive feedback, and we are very happy to hear that the reviewer is excited about this research direction. We address the questions below.
>
> **Mixer:** We agree, it is unclear how the token mixing (attention vs MLP) influences the inductive bias. While [1] does argue that Mixers have less inductive bias, we will highlight that this is debatable.
>
> **Compute vs samples:** We were referring to lines 257-267, where we more closely dissect how compute-optimal MLPs trade-off parameter count and number of samples. We find that for a given level of compute $C$, an optimal MLP invests $\propto C^{0.35}$ into parameter count, and $\propto C^{0.65}$ into sample size. This is in contrast to other architectures like transformers, where both quantities scale roughly the same as $\propto C^{0.5}$  (see [7]).
>
> **Normalization:** We have experimented with both Layer and BatchNorm and found that LayerNorm performs slightly better when pre-training on IN1k (see Fig. 7 in Appendix). We have extended this result now to IN21k, see Table 4 in pdf. Concretely, we pre-train a B-6/Wi-1024 B-MLP with BatchNorm on IN21k and fine-tune on all datasets. We again find that LayerNorm slightly outperforms BatchNorm across all the tasks.
> We have also decided to focus on LayerNorm due to its simplicity (no discrepancy for train and inference), as well as to be compatible with more modern architectures like ViT. To our knowledge, ViTs rely on LayerNorm largely due to the fact that there is no batch dependence, allowing more efficient training on multiple GPUs.
>
> **Bottleneck:** This is a great question. We managed to significantly improve performance of V-MLPs (the training protocol was sub-optimal before). We observe that they become more competitive with B-MLPs when training from scratch, thus downplaying the importance of the bottleneck layer (see Table 1). We also pre-trained a V-MLP of width 2048 and depth 6 on IN21k and compared it with a B-6/Wi-1024 B-MLP (both models around 70M params) in terms of fine-tuning (see Table 3). We observe that B-MLPs still generalize better but V-MLPs also reach strong performance. We further find that V-MLPs also exhibit very similar power-laws (see Fig. 1, left) with a slightly worse slope than B-MLPs (0.22 vs 0.25), as expected. We think this further strengthens our claims regarding inductive bias and also makes our findings more directly compatible with theory settings.
>
> **Label Smoothing:** Upstream label smoothing does help performance but is not crucial. We have added results for more smoothing amounts in Table 2. We see that across all tasks, performance improves with stronger smoothing upstream, but only moderately. Thus, we are very confident that the scaling laws still apply even for no smoothing, albeit with a slightly weaker slope.
> Strong amounts of smoothing have been used previously, for instance [2] use smoothing of 0.2, notably also for IN21k. We believe this is effective as samples from IN21k often contain multiple objects, whereas only one ground truth label is provided, as also argued in [2]. Label smoothing in general has seen a resurgence, see e.g. [3, 4, 5].
>
> **Train accuracies:** In Table 5 (pdf) we show training accuracies both on the augmented and un-augmented set as well as test accuracies on IN21k. A moderately large B-MLP trained without augmentations easily overfits IN21k, leading to a large generalization gap. The training accuracies on the augmented set are small largely due to MixUp. We have also added a ViT B/8 baseline to highlight that “underfitting” the original training data when training with augmentations is typical.
>
> **Validation:** For downstream tasks we do not use a validation set but rather fix the number of epochs such that training accuracy saturates. We do not tune hyperparameters but choose a configuration that allows training across all tasks.  For upstream we use a validation set to track progress. We experiment with different hyper-parameters upstream and have stopped runs early in training if upstream validation accuracy does not improve. Tuning hyper-parameters upstream with this strategy (if at all) is common in previous papers investigating scaling laws see e.g. [6, 7, 8].
>
> **V-MLP:**  We did not report later epochs for the V-MLP since it already fits the training data perfectly (also applies for the improved version), and later epochs only deteriorate the performance. We have added more epochs in Table 1. The ResNet was trained for 100 epochs.
>
> **Stop wrt train accuracy:** For downstream training as well as training from scratch for small datasets, we indeed rely on training accuracy/loss (and their saturation) to determine the length of training. When employing augmentations, the models improve more slowly, hence we choose to also evaluate at later epochs.
>
> **Batch size:** We apologize, the plot depicts downstream test error as a function of pre-training batch size, not accuracy.
>
> **Augmentation:** We agree that we were not precise. Augmentations encode invariances in the data (e.g. rotating an image does not affect its content). By training on such augmentations, we try to instill symmetries into the network, and thus arguably add some inductive bias implicitly. We believe that also the noisy nature of augmentations helps with overfitting.
> \
> \
> We will include these results in the text and thank the reviewer again for all the questions. We truly believe that addressing them helped improve our work significantly.
>
> \
> [1] MLP-Mixer: An all-MLP Architecture for Vision, Tolstikhin et al.
>
> [2] ImageNet-21K Pretraining for the Masses, Ridnik et al.
>
> [3] Training data-efficient image transformers, Touvron et al.
>
> [4] ResNet strikes back, Wightman et al.
>
> [5] How to train your ViT? Steiner et al.
>
> [6] Scaling laws for large language models, Kaplan et al.
>
> [7] Training Compute-Optimal Large Language Models, Hoffmann et al
>
> [8] Scaling Laws for Autoregressive Generative Modeling, Henighan et al.

---

> > ### Author Response · Authors · 2023-08-21
> >
> > Dear reviewer **xaDV**,
> >
> > As the discussion phase ends today, we will not be able to further clarify potential additional questions you may have. We would be very grateful if you could respond to our rebuttal, which includes many experiments aimed at addressing your questions. This would provide us with an opportunity to further engage with your concerns and potentially improve our work!
> > \
> > \
> > Best,\
> > Authors

---

### Author Rebuttal · Authors · 2023-08-08

We thank all the reviewers for their valuable feedback. We reply to each reviewer’s questions in the individual responses and have added tables and figures in the attached rebuttal pdf which we reference and explain in the responses. For convenience we also provide a detailed summary below.
\
\
Due to space constraints, we had to compress notations. For datasets we denote CIFAR10 → C-10, CIFAR100 → C-100, TinyImageNet → T-IN and ImageNet → IN. We further use V-MLP for vanilla MLP and B-MLP for Bottleneck MLP.  We also denote upstream by $\uparrow$ as well as downstream by $\downarrow$. \
To denote the architecture of V-MLPs, we slightly abuse notation and also denote by B-L/Wi-M a vanilla MLP of depth $L$ and width $M$. Notice that we used the same notation for Bottleneck MLPs but a block there is slightly different (i.e. first maps to $4 \times M$ and then to $M$). We refer to the main text for the more precise description.
\
\
To summarize the results in the pdf, we have found the following.

1. We managed to improve performance of the **Vanilla MLPs** on small datasets (the training protocol was sub-optimal before) making them more competitive with Bottleneck-MLPs when training from scratch (see Table 1 in rebuttal pdf). To further investigate the scaling behaviour of Vanilla MLPs, we pre-trained them on ImageNet21k and fine-tuned them on several tasks (see Table 3 in rebuttal pdf). Comparing against a similar sized Bottleneck-MLP, we see that the B-MLP still performs better, but the V-MLP also achieves very strong performance. To complete the picture, we produce the same scaling plot as done for B-MLPs (Figure 4 in paper) in Figure 1 (left) of the rebuttal pdf, i.e. we pre-train variously-sized V-MLPs on several subsets of ImageNet21k and plot the amount of compute (in FLOPs) against the test error of linear evaluations on CIFAR-100. We observe that V-MLPs exhibit the same power-law behaviour as B-MLPs with a slightly weaker slope (0.22 vs 0.25). While the bottleneck layer thus helps in terms of absolute performance, it is not essential to the scaling behaviour. We think this further strengthens our claims regarding inductive bias and also makes our findings more directly compatible with theory settings.

2. We ablated the effect of **label smoothing** upstream by pre-training B-MLPs on ImageNet21k with various smoothing amounts and subsequently fine-tuning them on several datasets. We found that smoothing does help performance but only very marginally so (see Table 2 in rebuttal pdf), e.g. a B-MLP B-6/Wi-1024 without any smoothing performs almost on-par compared to smoothing with a value of 0.3 (90% vs 89.2% on CIFAR-10). Smoothing is thus not an essential component to our findings.


3. We extended the results of Figure 7 in the Appendix and **compared LayerNorm and BatchNorm** by pre-training B-MLPs on ImageNet21k and subsequently fine-tuning them on several tasks (see Table 4 in rebuttal pdf). We confirm that LayerNorm has a slight edge over BatchNorm in terms of downstream performance, but only marginally so (e.g. 90.0% vs 89.4% on CIFAR-10).


4. We have added **upstream training and test accuracies** on ImageNet21k in Table 5 of the rebuttal pdf. We evaluate training accuracy both for augmented and un-augmented versions. We find that a moderately large B-MLP B/6-Wi-1024 easily overfits the training set if trained without augmentations, leading to strong overfitting behaviour. Overfitting is reduced by employing data augmentation, which serves as a regularizer in this setting.

5. In Figure 1 (right) we show how MLPs are robust to **different resolutions** when the embedding layer is adequately resized, adapting the approach of FlexiViT [1]. We take a B-12/Wi-1024 B-MLP which is pre-trained on ImageNet21k at a resolution of $64 \times 64$ and fine-tune it on ImageNet1k at a resolution of $64 \times 64$. We then evaluate test performance on ImageNet1k for resolutions $\{16, 32, 48, 64, 96, 128\}$ while resizing the embedding layer as outlined in [1] in Section 3.4 (without any further training). We observe that test performance remains stable for all resolutions and slowly starts deteriorating only for the smallest ones.

We again thank the reviewers for the very helpful feedback and we truly believe that addressing the questions significantly improved our work.

Finally, we agree with the reviewers and we will add a separate **Limitations section**.
\
\
[1] FlexiViT: One Model for All Patch Sizes, Beyer et al.

---

### Decision · Program_Chairs · 2023-09-21

**Decision:**

Accept (poster)

**Comment:**

Reviewers find this paper studying interesting and under-explored questions about scaling behavior of simple MLP models and its similarity to popular architectures (e.g. Transformers) behavior with scale. They find the paper results, though preliminary, insightful and encouraging warranting further studies.

Reviewers raised some valid criticisms of the paper. 1) Some claims around MLPs having less inductive bias are questionable as Transformers are  also universal approximator function class. 2) Evaluations are limited to image classification and small scale datasets/models. 3) Low accuracies on ImageNet.

While I agree with limitations of this paper, I think the direction is interesting and the paper takes a good enough step towards it. Hence I suggest acceptance. It is important that authors update the paper addressing the limitations as suggested by the reviewers and toning down some of the claims.